# Combination Therapy Approach to Overcome the Resistance to PI3K Pathway Inhibitors in Gynecological Cancers

**DOI:** 10.3390/cells13121064

**Published:** 2024-06-19

**Authors:** Kristen R. Ibanez, Tzu-Ting Huang, Jung-Min Lee

**Affiliations:** Women’s Malignancies Branch, Center for Cancer Research, National Cancer Institute, National Institutes of Health, Bethesda, MD 20892, USA; kr892361@ucf.edu (K.R.I.); leej6@mail.nih.gov (J.-M.L.)

**Keywords:** PI3K/AKT/mTOR inhibitors, resistance mechanisms, combination treatment, targeted therapy, gynecological cancer

## Abstract

The PI3K signaling pathway plays an essential role in cancer cell proliferation and survival. PI3K pathway inhibitors are now FDA-approved as a single agent treatment or in combination for solid tumors such as renal cell carcinoma or breast cancer. However, despite the high prevalence of PI3K pathway alterations in gynecological cancers and promising preclinical activity in endometrial and ovarian cancer models, PI3K pathway inhibitors showed limited clinical activity in gynecological cancers. In this review, we provide an overview on resistance mechanisms against PI3K pathway inhibitors that limit their use in gynecological malignancies, including genetic alterations that reactivate the PI3K pathway such as *PIK3CA* mutations and *PTEN* loss, compensatory signaling pathway activation, and feedback loops causing the reactivation of the PI3K signaling pathway. We also discuss the successes and limitations of recent clinical trials aiming to address such resistance mechanisms through combination therapies.

## 1. Introduction

The phosphoinositide 3-kinase (PI3K) pathway is essential for regulating cell metabolism, growth, and proliferation under normal physiological conditions. However, the aberrant activation of this pathway is a common oncogenic driver in various solid tumors [1]. Targeting the PI3K pathway has become a critical focus in cancer therapy, leading to the development and FDA approval of several inhibitors over the past two decades. Initially, mTOR inhibitors such as everolimus and temsirolimus were approved for treating renal cell carcinoma, followed by approvals for other solid tumors [2]. More recently, alpelisib and capivasertib, PI3Kα and AKT inhibitors, respectively, have been approved in combination with fulvestrant for hormone receptor-positive, HER2-negative metastatic breast cancer [3].

Despite the frequent dysregulation of the PI3K pathway in gynecological cancers such as endometrial cancer (EC) and ovarian carcinoma (OC), PI3K pathway inhibitors have yet to show sufficient clinical efficacy for approval in these settings [4]. Early clinical trials investigating PI3K pathway inhibitor monotherapies in EC and OC have demonstrated only modest activity, failing to replicate the promising results observed in preclinical studies [4]. This discrepancy is likely due to various resistance mechanisms that limit the effectiveness of PI3K pathway inhibitors in these tumors. The presence of de novo resistance mechanisms, where tumors show no initial clinical response due to intrinsic refractoriness or rapid adaptation to PI3K inhibition, may obscure treatment effects or necessitate higher doses to achieve efficacy, potentially leading to intolerable side effects [3]. In addition, resistance mechanisms can develop after the initiation of treatment, described as acquired resistance mechanisms [3]. 

This review aims to provide an overview of the resistance mechanisms associated with PI3K pathway inhibitors in gynecological cancers. It will also explore combination therapy strategies designed to overcome these resistance mechanisms and enhance the clinical utility of this important drug class in gynecological malignancies.

## 2. Brief Overview of PI3K Pathway and the FDA-Approved PI3K Inhibitors

The PI3K signaling cascade begins with the activation of receptor tyrosine kinases (RTKs) or G protein-coupled receptors (GPCRs) upon the binding of extracellular ligands, such as growth factors [1]. This activation triggers PI3K to convert phosphatidylinositol-4,5-bisphosphate (PIP2) into phosphatidylinositol-3,4,5-triphosphate (PIP3), a second messenger that recruits AKT to the cell membrane [3]. AKT is then activated by the mTOR complex 2 (mTORC2), which subsequently leads to the activation of mTOR complex 1 (mTORC1) [5]. In contrast, the tumor suppressor PTEN dephosphorylates PIP3 back to PIP2, counteracting PI3K activation [3]. The loss of PTEN function or the constitutive activation of upstream factors (RTKs, PI3K, AKT, and, in rare cases, mTOR) can result in the hyperactivation of the PI3K pathway, contributing to the development and progression of various malignancies [1].

Due to the central role of the PI3K pathway in cell growth regulation, many inhibitors targeting different components of the PI3K/AKT/mTOR pathway have been developed. Figure 1 illustrates the PI3K pathway and the sites of action for selected PI3K pathway inhibitors under clinical investigation for advanced solid tumors. The newest inhibitors in development focus on selectively targeting specific mutations of *PIK3CA* that frequently affect class 1A PI3K, such as the H1047R residue in the kinase domain [6,7]. Table 1 summarizes FDA-approved PI3K pathway inhibitors and their approved indications.

### 2.1. Frequency of PI3K Pathway Alterations in Gynecological Cancers

In EC, the dysregulation of the PI3K pathway is a central factor in tumor initiation, progression, and therapeutic response [8]. PI3K pathway mutations are commonly observed across the four molecular groups of EC classified by The Cancer Genome Atlas (TCGA): ultramutated POLE (71% *PIK3CA*, 65% *PIK3R1*, 93% *PTEN*), microsatellite instability (MSI) high (*PIK3CA* 55%, *PIK3R1* 40%, *PTEN* 88%), copy number low (CNL; *PIK3CA* 53%, *PIK3R1* 34%, *PTEN* 77%), and copy number high (CNH; *PIK3CA* 53%, *PIK3R1* 13%, *PTEN* 15%) [9]. Notably, EC shows the highest frequency of PI3K pathway alterations (89%, including *PIK3CA*, *PIK3R1*, *PTEN*, *AKT1/2/3*, and *MTOR*) than any other cancers based on TCGA data (www.cbioportal.org [10], accessed on 29 May 2024). 

In OC, *PIK3CA* amplification occurs in 19% of cases, while the frequency of *PIK3R1* alterations is rare (~3%) [11,12]. Aberrations in downstream effectors, such as *AKT1/2/3* and *MTOR*, are also less frequent, with 14% and approximately 4% of OC cases harboring alterations in these genes, respectively [12]. Cervical cancer, although less extensively studied, also exhibits PI3K pathway alterations. *PIK3CA* mutations occur in 38% of cases, and *PIK3R1* alterations are present in 3% [13]. Additionally, 10% of cervical tumors show alterations in *AKT1/2/3* [11], and approximately 4% harbor alterations in *MTOR* [13]. 

Overall, gynecological cancers have a high prevalence of PI3K pathway alterations, thus establishing the rationale for PI3K pathway inhibitors in the treatment of gynecological cancers.

### 2.2. PI3K Pathway Monotherapy Efficacy in Gynecological Cancers

Current studies on PI3K pathway inhibitors often focus on patients with mutations in the PI3K pathway, such as *PIK3CA*, *PIK3R1*, *AKT* mutations, or *PTEN* loss, as biomarkers for patient selection [14]. However, early clinical trials using PI3K pathway inhibitor monotherapy such as buparlisib in heavily pretreated gynecological cancers have shown limited efficacy, with overall response rates (ORR) generally below 10% [4], except for a few phase II trials demonstrating moderate efficacy (ORR 10–26%) in recurrent or persistent EC [15,16]. Even when selecting EC patients with mutations in *PTEN*, *PIK3CA*, *PIK3R1*, *AKT*, or *MTOR*, the ORR remained around 16% [17], suggesting the involvement of resistance mechanisms beyond simple hotspot mutations of the PI3K pathway. Therefore, it is necessary to further investigate these resistance mechanisms to improve therapeutic outcomes.

## 3. Mechanisms of Resistance

Previous trials have described genetic alterations associated with potential clinical PI3K monotherapy resistance in gynecological tumors and are detailed in Table 2. However, truly conclusive data for de novo and acquired resistance biomarkers are not available due to the exploratory and retrospective nature of these studies as well as the lack of advanced-phase clinical trials in these cancer types. This section aims to review existing evidence on the most frequently characterized resistance mechanisms to PI3K inhibitors.

### 3.1. De Novo Resistance

#### 3.1.1. Primary Amplification/Mutations in *PIK3CA*

*PIK3CA* amplification and mutations are common in gynecological cancers, yet their role as biomarkers remains debated. The strongest evidence can be found with the phase III studies of alpelisib (SOLAR-1 trial; ORR 26.6% vs. 12.8% in *PIK3CA*-altered vs. *PIK3CA* wild-type tumors) and capivasertib (CAPItello-291 trial; ORR 28.8% vs. 22.9% in *AKT*-altered tumors vs. overall population) in hormone receptor-positive, HER2-negative metastatic breast cancer. While some clinical activity was observed in *PIK3CA* wild-type tumors, these trials only demonstrated statistically significant improvement in progression-free survival (PFS) in *PIK3CA*-mutated populations [22,23,24], leading to FDA approval for these agents solely in *PIK3CA*-mutated patients (also *AKT1/PTEN*-mutated patients for treatment with capivasertib). 

However, despite these data that encourage the use of PI3K pathway alterations as a biomarker for response, other studies have proposed *PIK3CA* mutations as a resistance mechanism [25]. For instance, Huw et al. reported that breast cancer cells with *PIK3CA* amplification or mutations exhibited resistance to a PI3Kα/δ inhibitor, pictilisib, due to the high amplification of the mutant *PIK3CA* exon 20 p.H1047R locus, leading to increased AKT phosphorylation [25]. Another important consideration includes the presence of multiple *PIK3CA* mutations prior to treatment, as 27% of ECs harbor more than one *PIK3CA* mutations, a feature more common in EC than any other tumor type [26]. Some authors suggest that multiple clonal *PIK3CA* mutations occurring in cis respond better to PI3K inhibition than a single *PIK3CA* mutation or multiple subclonal *PIK3CA* mutations occurring in trans [26]. In line with preclinical findings, secondary analyses from the SANDPIPER trial showed lower ORR (7.1% vs. 41%) and worse PFS (median PFS: 5.1 months vs. 7.6 months) in patients with multiple subclonal *PIK3CA* mutations compared to multiple clonal *PIK3CA* mutations in metastatic breast cancer treated with taselisib and fulvestrant [27].

#### 3.1.2. Loss of Negative Regulator *PTEN*

PTEN is a key negative regulator of the PI3K pathway, and its loss increases PI3K pathway output [28]. While *PTEN* loss may not confer resistance to pan-PI3K inhibitors like pictilisib [29], it contributes to resistance against PI3Kα-specific inhibitors, such as alpelisib [30]. *PTEN*-null tumors primarily rely on PI3Kβ rather than PI3Kα for growth [31,32]. Consequently, *PTEN*-null breast cancer models show resistance to PI3Kα inhibitors but remain sensitive to pan-PI3K inhibitors [30], suggesting *PTEN* loss as a biomarker is more relevant for PI3Kα-specific inhibitors.

### 3.2. Acquired Resistance Due to PI3K Pathway Alterations

#### 3.2.1. Secondary Mutations in *PIK3CA* and Acquired Activating *AKT* Mutations

As mentioned earlier, new PI3Kα inhibitors are under development, and while encouraging, these inhibitors have also already shown evidence of acquired resistance. Inavolisib is a PI3Kα inhibitor currently under investigation in a phase III trial evaluating its combination with palbociclib and fulvestrant as a first-line treatment for *PIK3CA*-mutated, hormone receptor-positive, HER2-negative, endocrine-resistant, locally advanced or metastatic breast cancer (NCT04191499). Preliminary data showed the inavolisib combination improved PFS compared to palbociclib and fulvestrant alone (15.0 months vs. 7.3 months) [33,34]. However, secondary *PIK3CA* mutations and acquired activating *AKT* mutations have been suggested as mechanisms of resistance. For instance, the *PIK3CA* Q859 and W780 mutations, and *AKT* E17K/Q79K mutations, confer resistance to inavolisib and alpelisib in breast cancer patients with baseline *PIK3CA* mutations (H1047R, E542K/E545K) [35]. This study provides further indication that some *PIK3CA* mutations may actually drive resistance, rather than sensitivity, to PI3Kα inhibitors.

#### 3.2.2. Acquired Activating *PIK3CB* Mutations in *PTEN*-Loss Tumors

Apart from *PIK3CA*, an acquired activating *PIK3CB* (D1067Y) mutation has been reported to cause resistance to the pan-PI3K inhibitor GDC-0941 in a *PTEN*-loss context [36]. This mutation activates p110β, maintaining PI3K signaling despite PI3K inhibition [36]. Such tumors may develop *PIK3CB* mutations or favor rare clones during treatment. Despite this, the stable expression of mutant *PIK3CB* variants can be overcome by AKT or mTORC1/2 inhibitors in breast cancer cell lines [36]. Therefore, screening for *PIK3CB* mutations in PTEN-deficient patients progressing on PI3K inhibitors may be warranted. Additionally, exploring therapeutic alternatives targeting AKT or mTORC1/2 may benefit PTEN-deficient patients resistant to PI3K inhibitors.

#### 3.2.3. Activation of Upstream Effectors and Insulin Feedback

Resistance to PI3K pathway inhibitors often arises from signaling feedback loops. One notable mechanism involves the upregulation of upstream effectors like insulin-like growth factor receptor (IGF-1R) when PI3K is inhibited, leading to insulin feedback and transient hyperglycemia (Figure 2). This feedback loop maintains PI3K pathway activity, limiting the effectiveness of PI3K inhibitors [4,37].

Hyperglycemia-related dose-limiting toxicities pose challenges in clinical settings [38,39]. Strategies like insulin feedback suppression with SGLT-2 inhibitors or a ketogenic diet are being explored to improve PI3K pathway inhibitor sensitivity. These approaches improve PI3K pathway inhibitor sensitivity to alpelisib, taselisib, and buparlisib in murine models [40]. However, metformin alone did not rescue insulin-suppressed buparlisib activity [40]. Therapeutic benefits have also been observed in patient-derived xenograft (PDX) models of advanced EC with a combination of buparlisib and a ketogenic diet [40].

Other strategies are currently in development to reduce on-target effects on metabolism from wild-type PI3K inhibition. PI3Kα mutant-selective inhibitors (e.g., STX-4786 [6], RLY-26087 [7], and LOXO-783 [41]) targeting hotspot mutations like H1047R and E542K/E545K have been developed, although early phase clinical trials are still underway. Additionally, certain orthosteric PI3K inhibitors, such taselisib and the aforementioned inavolisib, degrade mutant p110α protein, enhancing cytotoxicity and allowing for lower effective doses [42].

### 3.3. Mechanisms of Acquired Resistance Outside of PI3K Pathway Alteration

In response to PI3K inhibition, cancer cells may engage in the compensatory upregulation of alternative pathways that support cell growth and survival, bypassing the effects of PI3K pathway inhibition (Figure 3). Two significant pathways involved are the Ras/Raf/mitogen-activated protein kinase (MEK)/extracellular signal-regulated kinase (ERK) pathway and the Wnt/β-catenin pathway. Additionally, other mechanisms such as cell cycle checkpoint activation also contribute to resistance.

#### 3.3.1. Ras/RAF/MEK/ERK Pathway Activation

In gynecological cancers, particularly OCs and ECs, the activation of the KRAS pathway and the presence of *KRAS* mutations are substantial factors contributing to both de novo and acquired resistance against PI3K pathway inhibitors [43] (Table 2). KRAS pathway activation can act as a compensatory mechanism when the PI3K pathway is inhibited, allowing cancer cells to maintain survival and growth [43,44]. For example, *KRAS* mutations, which are present in 45% of EC, 30% of OC, and 20% of cervical cancers, often lead to continuous signaling through the KRAS pathway, even in the absence of upstream growth factor stimulation [11].

Activating mutations such as G12D and G12V are of significant interest [45], as these mutations render KRAS constitutively active, driving continuous signaling through the KRAS pathway, even in the absence of upstream growth factor stimulation. While low-grade serous ovarian carcinomas (LGSOC) comprise a smaller proportion of OCs compared to high-grade serous ovarian carcinomas (HGSOC), LGSOC is characterized by more MAPK pathway mutations (30–60% of LGSOCs) [46]. 

Preclinical evidence has demonstrated the efficacy of combined PI3K pathway and MEK/ERK inhibition in OC models. For instance, dual inhibition using a dual PI3K/mTOR inhibitor PF-04691502 and a MEK inhibitor PD-0325901 showed in vivo efficacy in a genetically engineered mouse model of OC driven by *KRAS* G12D-activating mutation and *PTEN* deletion [47]. Synergistic cytotoxic effects have also been observed in HGSOC cells treated with the combination of the dual PI3K/mTOR inhibitor dactolisib and the ERK inhibitor SCH772984 [48]. Efforts to improve *KRAS*-specific targeting to decrease cytotoxicity with the combination are also ongoing, including the use of *KRAS* gene silencing [49] and agents with newer mechanisms of action such as the HRAS and RHEB inhibitor, tipifarnib [50].

#### 3.3.2. Wnt/β-Catenin Pathway Activation

The Wnt/β-catenin pathway is another compensatory route activated in response to PI3K pathway inhibition. The upregulation of Wnt pathway components, such as Wnt ligands, frizzled receptors, beta-catenin (*CTNNB1*), and porcupine (*PORCN*), allows cells to continue proliferating despite PI3K inhibition. For example, triple-negative breast cancer (TNBC) cell lines treated with buparlisib exhibited an eight-fold overexpression of *PORCN*, contributing to resistance [51]. Similarly, colon cancer cells with elevated levels of β-catenin, a key mediator of the Wnt pathway, have been observed to exhibit resistance to AKT inhibition [52].

Combining PI3K inhibitors with Wnt/β-catenin pathway inhibitors has shown promise. Synergy between PI3K inhibitors and porcupine inhibitors has been demonstrated in TNBC cell lines and pancreatic cancer xenografts [51,53]. Another compelling study demonstrates the reversal of resistance to pan-PI3K, PI3Kα-selective, dual PI3K/mTOR, and AKT inhibitors in colorectal cancer patient-derived organoids and PDX models using a Wnt/tankyrase inhibitor, NVP-TNKS656 [54]. This study further proposes the utilization of nuclear β-catenin and FOXO3A activity as potential biomarkers for response prediction, which may benefit the future development of clinical trials. No preclinical studies have explored combinations of PI3K pathway inhibitors and Wnt/β-catenin pathway inhibitors specifically in gynecological cancers. However, the incidence of *RNF43* mutations in EC (18%) and the fact that RNF43 inactivation leads to increased susceptibility to PORCN inhibitors (through upregulation of Wnt receptors on the cell membrane) [55] suggests a rationale for exploring this approach in EC. As our understanding of the interactions between these pathways continues to mature, continued research into these combination strategies may be worthwhile for future clinical investigation.

#### 3.3.3. Endocrine Response

Alterations in estrogen signaling can be linked to resistance against PI3K pathway inhibitors, particularly in estrogen receptor (ER)-positive breast cancer; however, whether this translates to gynecological cancers sensitive to hormones is unknown. The suppression of PI3K signaling in ER-positive, *PIK3CA*-mutated breast cancer cell lines resulted in the induction of ER-dependent transcriptional activity, which were enhanced by the addition of estradiol and suppressed using the anti-ER therapies fulvestrant and tamoxifen [56]. Similar to breast cancer, EC has a large proportion with high sensitivity to estrogen signaling, ~92% of type I and ~72% of type II ECs were ER-positive [57], particularly within the POLE, MSI, and CNL molecular subtypes [9], making them potential candidates for combined PI3K and endocrine therapy.

#### 3.3.4. Activation of Cell Cycle Checkpoint Pathways

Cell cycle checkpoint pathways, such as ATR/CHK1 and WEE1, are crucial for maintaining genomic stability by regulating DNA repair and replication stress [4]. The PI3K pathway intersects with these checkpoints, influencing cell survival and proliferation [4]. The activation of these pathways can contribute to resistance against PI3K inhibitors. 

Wu et al. have reported that WEE1 activation following PI3K inhibition can confer resistance in glioblastoma cells [58]. Supporting this, the combination of PI3K/AKT and ATR/CHK1 inhibitors enhances cell death by impairing DNA repair and inducing replication stress in drug-resistant HGSOC cell lines and mouse models [59,60]. Combining ATR inhibitors with PI3K pathway inhibitors shows promise, especially in patients with *ARID1A* mutations, which are common in ovarian clear cell carcinomas (50%), ovarian endometrioid carcinomas (30%), and ECs (35%) [61]. Notably, *ARID1A* deficiency is one of the most promising biomarkers of ATR inhibitor monotherapy in early phase trials due to increased reliance on the G2/M checkpoint and subsequent replication stress [62]. Preclinical evidence shows that *ARID1A* knockout in EC cells leads to the upregulation of the PI3K pathway [63], suggesting patients with *ARID1A* mutations may benefit from combined ATR and PI3K pathway inhibition.

Another pathway involves targeting cyclin-dependent kinase 4/6 (CDK4/6) inhibitors to reverse PI3K inhibitor resistance [64]. CDK4/6 inhibitors such as palbociclib and ribociclib have been approved and are utilized in clinical practice as standard of care with hormonal therapy for hormone receptor-positive/HER2-negative metastatic breast cancer [65], reducing hormone receptor-activated cell cycle progression and the modulation of the tumor microenvironment [66]. While a direct link between the PI3K pathway and CDK4/6 inhibition in other hormone receptor-sensitive cancers like EC is not yet identified, preclinical evidence shows the potential synergy of combining these two pathway inhibitors in hormone receptor-positive/HER2-negative breast cancer cell line models [67]. It is unknown whether this combination may additionally have clinical utility in other hormone receptor sensitive cancers such as EC.

## 4. Clinical Evidence of Combination Therapies with PI3K Pathway Inhibitors and Current Trials in Progress

Combination therapy is a feasible approach to counteract resistance mechanisms to PI3K inhibitors. This section discusses the clinical evidence of combination treatments aimed at overcoming resistance. Several combination therapy trials targeting resistance mechanisms have been conducted in solid tumor patients, including those with gynecological cancers (Table 3). While some combination therapy trials with PI3K pathway inhibitors focus on combinations to improve the efficacy of existing therapies such as platinum and taxane therapy [68,69,70] or poly(ADP-ribose) polymerase (PARP) inhibitors [71,72,73]; in this section, we will focus on combinations with specific mechanisms aimed at overcoming PI3K inhibitor resistance.

### 4.1. Dual PI3K Pathway Inhibition

#### 4.1.1. Targeting Multiple PI3K Components: PI3K, AKT, and mTOR

Combining agents targeting multiple nodes of the PI3K pathway, such as PI3K, AKT, and mTOR, aims to comprehensively suppress signaling and enhance the inhibition of downstream effectors. However, phase I trials of such combinations in advanced solid tumors have shown increased toxicity and low clinical activity (ORR < 10%), resulting in their discontinuation (Table 3) [69,74,75].

#### 4.1.2. Combination with RTK Inhibition of FGFR, VEGF, and IGF-1

Combining PI3K pathway inhibitors with receptor tyrosine kinase (RTK) inhibitors, such as FGFR, VEGF, and IGF-1 inhibitors, has shown limited success due to high toxicity and low efficacy. Hyman et al. investigated the combination of the PI3Kα-selective inhibitor alpelisib and the pan-FGFR kinase inhibitor infigratinib in a phase I trial in advanced solid tumors with *PIK3CA* mutations with or without *FGFR* mutations [78]. This combination displayed similar tolerance compared to single agent activity, although the percentage of patients requiring a dose interruption was high (71% alpelisib and 60% infigratinib). 

Despite the involvement of IGF-IR in this pathway, preliminary clinical investigations into combination therapy with the IGF-1 receptor inhibitor dalotuzumab with AKT and mTOR inhibitors displayed limited activity. Brana et al. reported that in a phase I trial of advanced solid tumors, dalotuzumab with MK-2206, an AKT-inhibitor, showed no preliminary activity with an ORR of 0%. Similarly, these investigators also reported on a combination of dalotuzumab with ridaforolimus, a mTORC1-selective inhibitor, again showing no preliminary activity with an ORR of 0% in OCs [77]. Direct RTK inhibition within these combinations has shown limited activity and substantial toxicity, diminishing initial hopes for success. However, ongoing clinical trials are exploring alternative strategies to avoid RTK activation such as evading IGF-1 receptor stimulation by utilizing other agents that result in insulin suppression.

#### 4.1.3. Combination with Insulin Suppression

Strategies to avoid insulin feedback, such as using SGLT-2 inhibitors, AMPK activator metformin, and ketogenic diets, are being investigated to improve PI3K inhibitor efficacy. Trials investigating these combinations largely target EC, with one example being a phase II clinical trial that is actively recruiting participants to assess the combination of copanlisib with a ketogenic diet in relapsed or refractory ECs harboring *PIK3CA* or *PTEN* mutations (NCT04750941). Notably, metformin in particular is under investigation for use in EC outside of combination therapy investigations in a number of clinical trials due to additional mechanisms outside of the IGF-I/PI3K/mTOR pathway that are believed to play a role in the tumorigenesis of EC [92]. Given the improved tolerability profile of some of these agents, triplet combination therapies involving PI3K inhibition and insulin suppression are also under investigation, including a phase II trial investigating everolimus, letrozole and metformin combination in advanced ECs (NCT01797523) and a phase I trial investigating serabelisib, nab-paclitaxel, and an insulin-suppressing diet in advanced solid tumors with *PIK3CA* mutations (NCT05300048).

### 4.2. Combination with MEK Inhibitors

The inhibition of the Ras/Raf/MEK/ERK pathway, particularly with MEK inhibitors, can counteract compensatory survival mechanisms. MEK inhibitor monotherapy has been investigated in LGSOC in phase II trials. Here, *KRAS*-mutated tumors have been promising, with trametinib and binimetinib showing ORR of 50% and 44%, respectively, whereas the response of trametinib and binimetinib in non-*KRAS* mutated LGSOC was 8.3% and 19%, respectively [93,94]. While combination therapy of PI3K pathway inhibitors with MEK inhibitors has demonstrated efficacy in preclinical studies, translation into clinical studies has proven difficult, as it has been mainly restricted by DLTs. However, preliminary activity in a phase Ib trial with the MEK1/2 inhibitor trametinib and pan-class I PI3K inhibitor buparlisib appeared to display promising clinical activity in *KRAS*-mutated OCs (29% ORR, 6/21) in comparison to other solid cancers (1% ORR, 1/92) [81]. This led to further studies in OC, such as Arend et al. initiating a phase II trial of voxtalisib and pimasertib in LGSOC [84]. Although there were no differences in treatment-related ≥G3 adverse events between the pimasertib (84.4% of patients) and the pimasertib + voxtalisib arm (81.3%), this study was discontinued due to the high treatment discontinuation rate (~25% compared to anticipated 10%) related to overall toxicity, which included diarrhea, rashes, cardiac-related adverse events, and eye disorders such as macular or retinal detachment. The ORR from available data did not differ between the combination and pimasertib alone; in the combination treatment arm, 3/32 (ORR 9.4%) patients had a partial response, whereas 4/33 (ORR 12.1%) patients had a partial response in the pimasertib arm.

Overall, due to these observed toxicities in previous trials, further investigations of these combinations will need to balance activity with tolerability. Other methods are under investigation. A current phase I trial clinical trial is in progress, investigating the combination of everolimus and avutometinib for advanced solid tumors, which will be the first PI3K pathway and MEK pathway combination to target RAF/MEK rather than MEK1/2 (NCT02407509).

### 4.3. Combination with Endocrine Therapy

Combination therapy with endocrine therapy to reduce hormone sensitivity that may contribute to cell growth has proved an attractive method, particularly given that two PI3K pathway inhibitors approved for metastatic breast cancer are used in combination with fulvestrant, which works to both inhibit and degrade the ER. This combination in metastatic breast cancer is limited by some mutations in *ESR1* which encodes ERα, such as the *ESR1* Y537S or D538G mutation, which can activate estrogen signaling in the absence of estrogen and may promote PI3K pathway activation in breast cancer [95]. A study by Drouyer et al. investigated the incidence and clinical relevance of circulating *ESR1* in advanced ECs, finding 4/21 patients with *ESR1* mutations; however, none appeared to be clinically relevant or related to the development of hormone therapy resistance [96].

In gynecological cancers, this approach has been of interest particularly in the case of recurrent EC. A phase II study investigating the combination of everolimus and letrozole reported their primary endpoint of an ORR of 22% in persistent or recurrent EC and their secondary endpoints demonstrated improved PFS compared to hormone therapy alone in patients that were chemo-naïve [88]. Chemo-naïve patients demonstrated a 28-month median PFS; prior chemotherapy patients had a 4-month median PFS. On medroxyprogesterone acetate and tamoxifen combination therapy, patients without prior therapy had a 5-month median PFS; those with prior chemotherapy demonstrated a 3-month PFS [88]. Further phase II trials are investigating PI3K pathway inhibition in combination with various hormone therapies in EC (NCT02397083, NCT04486352, NCT05538897). 

Combination treatment in OC has been less successful. In a phase II trial of 20 ER+ HGSOC patients, Colon-Otero et al. demonstrated the modest efficacy of an everolimus and letrozole combination in relapsed patients with a reported ORR of 16% [87]. While no trials are currently underway that are investigating further combination in solely OC, a phase II trial is currently ongoing, investigating the efficacy of PI3Kα/δ-selective inhibitor copanlisib with the ER antagonist fulvestrant on ER-positive and/or progesterone receptor (PR)-positive OC, EC, and breast cancers with PI3K (*PIK3CA*, *PIK3R1*) and/or *PTEN* alterations (NCT05082025). Thus, while combination therapy involving PI3K pathway inhibitors and hormone therapies shows promising results in EC, there remains less enthusiasm for its effectiveness in OC.

### 4.4. Combination with Cell Cycle Checkpoint Inhibition

Combining PI3K inhibitors with cell cycle checkpoint inhibitors, such as CHK1 and ATR inhibitors, has shown potential but also significant toxicity in current clinical trials. A phase Ib trial tested a dual PI3K/mTOR inhibitor (samotolisib) and a CHK1 inhibitor (prexasertib) in 53 patients with advanced solid tumors [90]. The combination showed some preliminary clinical activity in a heavily pretreated population with an ORR of 15.3% but had difficulty with implementation due to toxicity [90]. Authors originally hoped that non-overlapping toxicities with the CHK1 inhibitor (mainly causing hematological effects) and PI3K/mTOR inhibition (primarily hyperglycemia and gastrointestinal effects) and a reduction in dose level would be tolerable. However, grade 3/4 adverse events remained common including neutropenia, thrombocytopenia, anemia, nausea, and fatigue, with authors reporting aggressive treatment with G-CSF would be needed for this combination.

It is also important to note that as a monotherapy, CHK1 inhibitors have been observed to cause a high degree of hematological toxicity, particularly compared to other inhibitors that target replication stress such as ATR or WEE1 inhibitors [62]. While ATR inhibitors have shown less clinical efficacy than CHK1 inhibitors as monotherapy treatments, its superior tolerability profile may prove to be a better candidate for combination therapy. A phase Ib trial was initiated investigating the combination of copanlisib and an ATR inhibitor elimusertib in advanced solid tumors with PI3K pathway or DNA damage response mutations (NCT05010096). However, the study was subsequently withdrawn by the sponsor due to no participants enrolled in the trial. Further investigations are needed with these combinations.

An ongoing phase I clinical trial is investigating the combination of the CDK4/6 inhibitor palbociclib in combination with the PI3K/mTOR inhibitor gedatolisib in advanced solid tumors (NCT03065062). In part II of this study, the investigators aimed to focus on tumors with suspected PI3K-pathway dependence such as EC. It will be interesting to note whether this combination has increased efficacy specifically in hormone receptor-positive tumors. No results are yet available and are eagerly awaited.

## 5. Conclusions and Future Directions

PI3K pathway inhibitors have yet to be incorporated into the clinical practice for treating gynecological malignancies due to resistance mechanisms leading to low clinical efficacy in early phase trials. Addressing resistance mechanisms, including genetic alterations, alternative signaling pathways, and feedback loops, may be the key in expanding the scope and efficacy of these inhibitors. Emerging research focuses on new therapeutic targets and combination strategies to overcome resistance. Notably, the Hippo/YAP1 pathway is activated when the PI3K pathway is inhibited, with YAP1 overexpression found in AKT inhibitor-resistant OC and EC cell lines [97]. Recently, targeting multiple modes of PI3K and its compensatory pathways like KRAS and YAP/YEAD has been suggested to benefit KRAS-driven tumors [98], but further investigation is needed in gynecological cancers. Additionally, significant challenges seen in attempted therapy combinations should be addressed, including intolerable toxicity [69,74,75] or a lack of improved clinical outcomes compared to existing treatments for recurrent tumors [71]. Collectively, the accumulated evidence on targeting PI3K pathway kinases and their associated pathways in gynecological cancers justify the continuing development of these inhibitors as a potential clinical combination therapy strategy.

## Figures and Tables

**Figure 1 cells-13-01064-f001:**
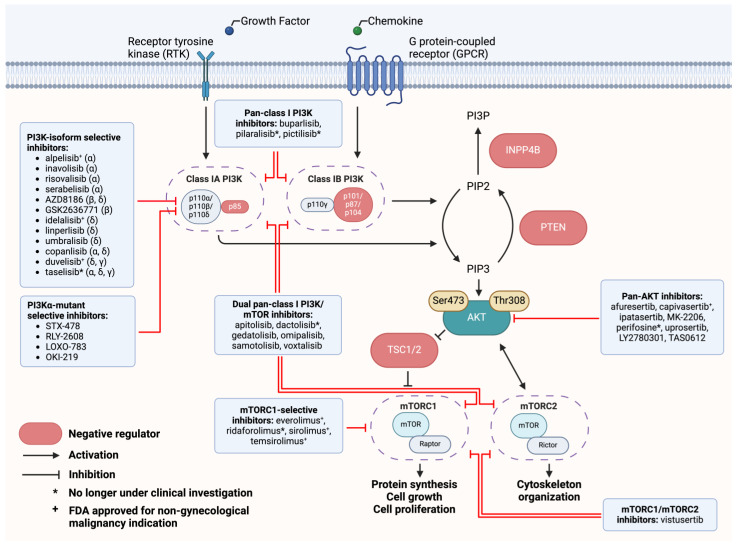
Target sites of selected PI3K pathway inhibitors. The phosphoinositide 3-kinase (PI3K) pathway comprises a series of molecular events beginning with the activation of receptor tyrosine kinases (RTKs) or G protein-coupled receptors (GCPRs) upon extracellular ligand binding by growth factor, which culminates in the activation of downstream effectors protein kinase B (AKT) and mammalian target of rapamycin (mTOR). Class I PI3K is a heterodimer composed of a regulatory subunit (p85, p101, p87, p84) as well as the catalytic subunit p110. The four p110 isoforms p110α, p110β, p110δ, and p110γ are encoded by the genes *PIK3CA*, *PIK3CB*, *PIK3CD*, and *PIK3CG*, respectively. Activated p110 catalyzes the conversion of phosphatidylinositol-4,5-bisphosphate (PIP2) into phosphatidylinositol-3,4,5-triphosphate (PIP3). PTEN plays an important role through negative regulation of the PI3K pathway, counteracting PI3K activation by dephosphorylating PIP3, and converting it back to PIP2. Inositol polyphosphate 4-phosphatase type II (INPP4B) converts PIP2 to phosphatidylinositol 3-phosphate (PI3P). PIP3 acts as a second messenger that promotes the translocation of AKT to the cell membrane. Once at the membrane, AKT is phosphorylated and activated by mTOR complex 2 (mTORC2). PI3K, AKT, and mTOR inhibitors exert their therapeutic effects by hindering their respective target along the PI3K pathway. * No longer in clinical development. ^+^ FDA-approved for treatment other than gynecological malignancy.

**Figure 2 cells-13-01064-f002:**
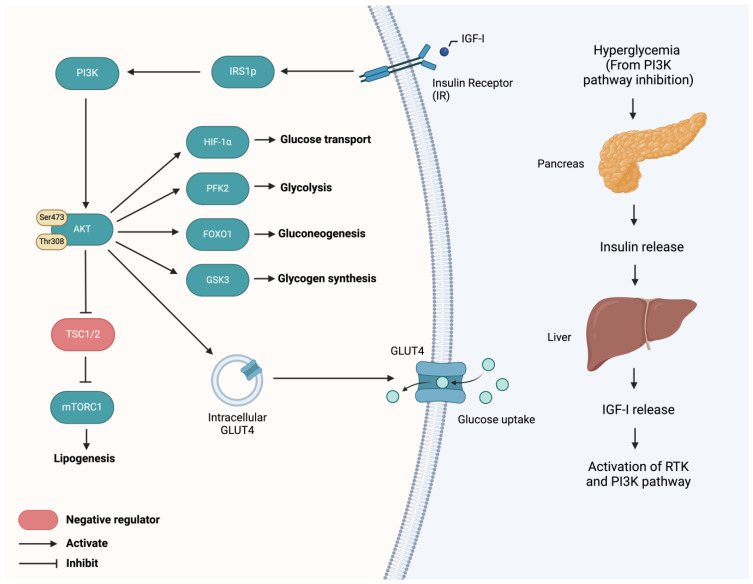
Role of PI3K pathway in glucose metabolism. Phosphoinositide 3-kinase (PI3K) inhibition additionally impedes normal PI3K pathway activation necessary for glucose metabolism initiated by insulin signaling. In normal glucose metabolism, insulin-like growth factor 1 (IGF-I) binds to insulin receptors (IR), leading to insulin receptor substrate 1 (IRS1) phosphorylation and subsequent activation of the PI3K pathway and downstream effects. Protein kinase B (AKT) downstream effects include consequences on glucose transport (GLUT4), glycogen synthesis (GSK3), glycolysis (PFK2), gluconeogenesis (FOXO1), lipogenesis (mTORC1) and glucose transporters (HIF-1α). In the presence of PI3K pathway inhibition, these effects are dysregulated with an overall effect of hyperglycemia. As a compensatory result of hyperglycemia, the pancreas increases insulin release, prompting a downstream increase in IGF-I production by the liver. This IGF-I continues to act on IR and lead to downstream effects.

**Figure 3 cells-13-01064-f003:**
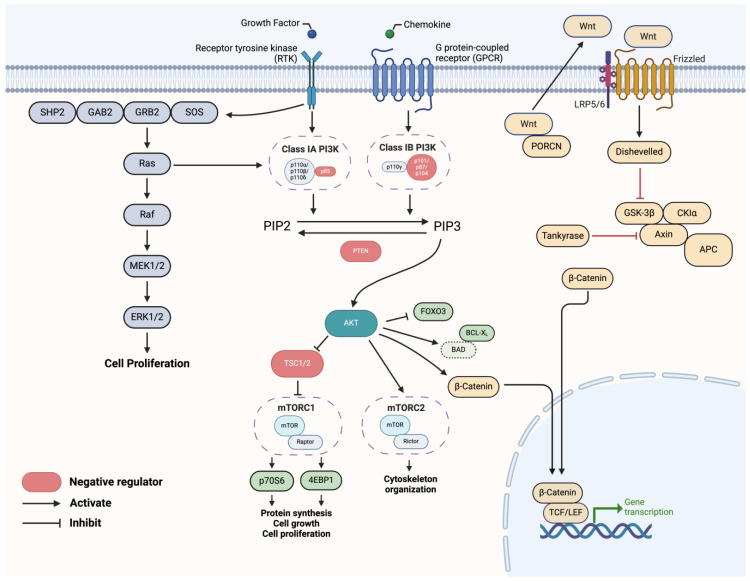
Parallel survival pathways in relation to the PI3K pathway. The phosphoinositide 3-kinase (PI3K)/protein kinase B (AKT)/mammalian target of rapamycin (mTOR) and Ras/RAF/mitogen-activated protein kinase (MEK)/extracellular signal-regulated kinase (ERK) pathways converge at Raf. Active RTKs expose phosphotyrosine residues for SHP2 docking. Phosphorylation of SHP2 enhances its activity and facilitates binding to the GRB2/SOS1 complex via GAB1, increasing RAS-GTP loading. PI3K inhibition increases Ras, an upstream activator of PI3K, in attempts to reactivate the PI3K pathway. When activated, Ras stimulates downstream effectors such as Raf and MEK, ultimately leading to the activation of the Ras pathway. This downstream signaling promotes cell division, survival, and proliferation. The PI3K pathway intersects with the Wnt/β-catenin pathway through AKT, a downstream effector of PI3K signaling. The Wnt/β-catenin pathway is a highly conserved signaling cascade involved in numerous biological processes. It is initiated by binding Wnt ligands to Frizzled receptors and LRP5/6 receptors on the cell surface. Without Wnt signaling, a destruction complex forms to degrade β-catenin. However, upon Wnt activation, β-catenin accumulates, enters the nucleus, and acts as a co-activator for transcription factors (TCF/LEF), leading to the expression of target genes involved in cell fate determination, proliferation, and tissue homeostasis. Acylation of Wnt by porcupine (PORCN) results in the secretion of Wnt ligands outside of the cell. Tankyrase interacts with and degrades AXIN via ubiquitin-mediated proteasomal degradation.

**Table 1 cells-13-01064-t001:** FDA-approved PI3K pathway inhibitors and indications.

Target	Drug	Initial FDA Approval Information	Current FDA-Approved Indications
PI3K	Idelalisib (PI3Kδ)	2014: Accelerated approval as a single-agent treatment for relapsed follicular lymphoma and small lymphocytic leukemia after two lines of therapy 2014: Regular approval in combination with rituximab for relapsed chronic lymphocytic leukemia in patients for whom rituximab alone would be considered appropriate therapy due to other comorbidities	Relapsed chronic lymphocytic leukemia, in combination with rituximab, in patients for whom rituximab alone would be considered appropriate therapy due to other co-morbidities 2022: FDA indication withdrawn for relapsed follicular lymphoma and small lymphocytic leukemia due to a higher incidence of fatal and/or serious adverse reactions in confirmatory trials
	Copanlisib (PI3Kα, δ)	2017: Accelerated approval as a single-agent treatment for relapsed follicular lymphoma who have received ≥ 2 prior systemic therapies	2024: FDA application withdrawn for third-line relapsed follicular lymphoma due to lack of benefit compared to standard of care in confirmatory trials
	Duvelisib (PI3Kδ, γ)	2018: Accelerated approval as a single agent for chronic lymphocytic leukemia, small lymphocytic lymphoma, and follicular lymphoma after ≥ 2 lines of therapy	Relapsed or refractory chronic lymphocytic leukemia or small lymphocytic lymphoma after at least two prior therapies 2021: FDA indication withdrawn for follicular lymphoma due to a higher incidence of fatal and/or serious adverse reactions in confirmatory trials
	Alpelisib (PI3Kα)	2019: Regular approval in combination with fulvestrant for postmenopausal women, and men, with hormone receptor-positive, human epidermal growth factor receptor 2 (HER2)-negative, *PIK3CA*-mutated, advanced, or metastatic breast cancer following progression on or after an endocrine-based regimen	Hormone receptor-positive, HER2-negative, *PIK3CA*-mutated, advanced, or metastatic breast cancer *PIK3CA*-related overgrowth syndrome
	Umbralisib (PI3Kδ)	2021: Accelerated approval as single agent for relapsed or refractory marginal zone lymphoma who have received ≥ 1 prior anti-CD20-based regimen, and relapsed or refractory follicular lymphoma who have received ≥ 3 prior lines of systemic therapy	2022: FDA indication withdrawn for marginal zone lymphoma and follicular lymphoma due to a higher incidence of fatal and/or serious adverse reactions in confirmatory trials
AKT	Capivasertib (pan-AKT isoform)	2023: Regular approval in combination with fulvestrant for hormone receptor-positive, HER2-negative, *PIK3CA*-mutated, advanced, or metastatic breast cancer with one or more *PIK3CA/AKT1/PTEN*-alterations, following progression on at least one endocrine-based regimen in the metastatic setting or recurrence on or within 12 months of completing adjuvant therapy	Hormone receptor -positive, HER2-negative, *PIK3CA/AKT1/PTEN*-mutated, advanced, or metastatic breast cancer
MTOR	Sirolimus (mTORC1)	1999: Regular approval as a single agent for prophylaxis of organ rejection in patients receiving renal transplants	Prophylaxis of organ rejection in patients receiving renal transplants Lymphangioleiomyomatosis Locally advanced unresectable or metastatic malignant perivascular epithelioid cell tumor Facial angiofibroma associated with tuberous sclerosis
	Temsirolimus (mTORC1)	2007: Regular approval as a single agent for advanced renal cell carcinoma	Advanced renal cell carcinoma
	Everolimus (mTORC1)	2009: Regular approval as single agent for advanced renal cell carcinoma after failure of either sunitinib or sorafenib	Postmenopausal women with advanced hormone receptor-positive, HER2 negative breast cancer in combination with exemestane after failure of treatment with letrozole or anastrozole Progressive neuroendocrine tumors of pancreatic origin and adults with progressive, well-differentiated, non-functional neuroendocrine tumors of gastrointestinal or lung origin that are unresectable, locally advanced or metastatic Advanced renal cell carcinoma after failure of treatment with sunitinib or sorafenib Renal angiomyolipoma and tuberous sclerosis complex, not requiring immediate surgery

**Table 2 cells-13-01064-t002:** Cases of genetic alterations associated with potential clinical resistance to PI3K pathway inhibitors in gynecological cancers.

Target	Drug	Type of Alteration	Genes Altered	De Novo/Acquired Resistance	Disease Setting
PI3K	Pilaralisib [18] (Pan-class I PI3K)	MAPK pathway-activating mutations	*KRAS* R68S	De novo	Advanced or recurrent EC
			*KRAS* G12D	De novo	
			*KRAS* G12V	De novo	
			*KRAS* G13D	De novo	
AKT	GSK2141795 [19] (Pan-AKT isoform)	MAPK pathway-activating mutations	*KRAS*, *NRAS* gain of function	De novo	Recurrent or persistent gynecological cancers including OC or EC
	Perifosine [20] (Pan-AKT isoform)	MAPK pathway-activating mutations	*KRAS* G13D	De novo	Recurrent or persistent gynecological cancers including OC, EC, or CC
			*HRAS* not specified	De novo	
		Wnt/β-catenin pathway alterations	*FBXW7* R465C	De novo	
			*FBXW7* R385C	De novo	
			*FBXW7* R226C	De novo	
MTOR	Everolimus [21] (mTORC1)	*PTEN* loss of function	*PTEN* c.728delT, p.Phe243fs	De novo	Advanced EC
			*PTEN* c.697C>T, p.Arg233X + c.950_953del, p.Val317fs	De novo	
		MAPK pathway-activating mutations	*KRAS* G12D	De novo	
			*KRAS* G12V	De novo	
Dual PI3K/ mTOR	Samotolisib [17] (Dual pan-class I PI3K/mTORC1/2)	MAPK pathway-activating mutations	*KRAS* G12S	Acquired	Advanced EC
		*FGFR2*-activating mutation	*FGFR2* P253R	Acquired	
		DNA damage repair alterations	*TP53* R273C	Acquired	
			*ATM* X823_splice	Acquired	
		*PIK3CA*-activating mutation	*PIK3CA* H1047L	Acquired	

Abbreviations: CC = cervical cancer, EC = endometrial cancer, OC = ovarian cancer.

**Table 3 cells-13-01064-t003:** Selected PI3K inhibitor combinatorial therapy clinical trials in gynecological cancers.

Drugs (Mechanism)	Phase	Eligible Patients	Common Adverse Events (>10%)	ORR (RECIST)
Dual PI3K pathway inhibition				
MK-2206 (Pan-AKT inhibitor) + ridaforolimus (mTORC1 inhibitor), Gupta 2015 [74]	I	Part A: Advanced solid tumors (*n* = 11) including EC (*n* = 1) Part B: Breast and prostate tumors with low *RAS* gene signature, high Ki67 index, or PTEN deficiency (*n* = 24)	G1/2: rash, stomatitis, diarrhea, fatigue, decreased appetite, asthenia, hyperglycemia, nausea, dry skin, thrombocytopenia, constipation, vomiting, hypertriglyceridemia G3/4: diarrhea	Part A: Not assessed Part B: ORR 8.3%, EC 0%
Dactolisib (Dual PI3K/mTOR inhibitor) + everolimus (mTORC1 inhibitor), Wise-Draper 2017 [75]	Ib	Advanced solid tumors (*n* = 19) including OC (*n* = 1), EC (*n* = 1)	G1/2: fatigue, fever, dehydration, anorexia, arthralgias, thrombocytopenia, anemia, AST elevation, ALT elevation, ALP elevation, rash, abdominal pain, nausea, vomiting, diarrhea, mucositis, acute renal failure G3/4: fatigue, dehydration, AST elevation, ALP elevation, anemia, mucositis, diarrhea	ORR 0% (*n* = 11)
Alpelisib (PI3Kα-selective inhibitor) + everolimus (mTORC1 inhibitor), Curigliano 2021 [76]	Ib	Doublet escalation phase: Advanced solid tumors (*n* = 13) including OC (*n* = 1), CC (*n* = 1), EC (*n* = 1)	G1/2: hyperglycemia, diarrhea, stomatitis, asthenia, decreased appetite, nausea, abdominal pain, increased blood creatinine, pyrexia, vomiting, weight decreased, anemia, arthralgia, ba-ck pain, dehydration, fatigue, non-cardiac chest pain, urinary tract infection, upper abdominal pain, acute kidney injury, aphthous ulcer, catarrh, constipation, cough, dry skin, dysgeusia, dyspnea, peripheral edema, skin ulcer G3/4: hyperglycemia, diarrhea, asthenia, hypertension, hyponatremia, increased lipase	Not assessed in doublet escalation phase
Receptor tyrosine kinase inhibition				
MK-2206 (Pan-AKT inhibitor) + dalotuzumab (IGF-1R inhibitor), Brana 2014 [77]	I	Part A: Advanced solid tumors (*n* = 24) including OC/PC/FC (*n* = 7)	G1/2: fatigue, decreased appetite, hyperglycemia, muscle spasm, dry skin, rash, rash maculopapular, dysgeusia, weight decreased, pyrexia, stomatitis G3/4: hyperglycemia	ORR 0%, OC/PC/FC 0%
Ridaforolimus (mTORC1 inhibitor) + dalotuzumab (IGF-1R inhibitor), Brana 2014 [77]	I	OC/PC/FC with RAS below 50th %, IGF1 expression above 75th % (*n* = 6)	G1/2: leukopenia, thrombocytopenia, dry eye, dry mouth, diarrhea, nausea, oral pain, stomatitis, vomiting, fatigue, mucosal inflammation, infusion-related reaction, decreased appetite, hypercholesterolemia, hyperglycemia, hypertriglyceridemia, pain in extremity, dysgeusia, cough, pneumonitis, dry skin, petechiae G3/4: abdominal pain, fatigue, decreased appetite, hyperglycemia, hypertriglyceridemia, hypophosphatemia, ECG T-wave inversion, muscle spasm	ORR 0%
Alpelisib (PI3Kα-selective inhibitor) + infigratinib (pan-FGFR kinase inhibitor), Hyman 2019 [78]	Ib	Advanced solid tumors with *PIK3CA* mutations ± *FGFR* alterations (*n* = 62) including EC (*n* = 6), OC (*n* = 5)	G1/2: diarrhea, decreased appetite, stomatitis, nausea, hyperphosphatemia, hyperglycemia, fatigue, dry mouth, vomiting, mucosal inflammation, blood creatinine increased, alopecia, asthenia, dry skin G3/4: hyperglycemia, stomatitis	ORR 9.7%, EC 0%, OC 16.7%
Temsirolimus (mTORC1 inhibitor) + cediranib (VEGFR inhibitor), Campos 2022 [79]	I	Gynecological cancers (*n* = 11) including EC (*n* = 4), OC/PC/FC (*n* = 4), CC (*n* = 3)	G1/2: diarrhea, nausea, vomiting, leukopenia, neutropenia, hypertriglyceridemia, elevated AST, elevated ALT, hypercholesterolemia, hyponatremia, proteinuria, abdominal pain, hypertension, mucositis, anorexia, elevated ALP, hyperglycemia, hypophosphatemia, hypokalemia, hypomagnesemia, hypothyroidism, epistaxis, fatigue, headache, rash, dry skin, alopecia, nutritional deficiency G3/4: hypertension, thrombocytopenia, thromboembolic event, edema, hypertriglyceridemia	ORR 0%
Insulin feedback suppression				
Temsirolimus (mTORC1 inhibitor) + metformin (biguanide derivative), Ahmed 2023 [80]	I	Advanced or recurrent EC (*n* = 40)	G1/2: hypertriglyceridemia, diarrhea, mucositis, anorexia, anemia	ORR 6%
Serabelisib (PI3Kα-selective inhibitor) + canagliflozin (SGLT2-inhibitor) (NCT04073680)	Ib/II	Advanced solid tumors with *PIK3CA* or *KRAS* mutations	Study status: Unknown status on CT.gov as of May 2020	
Copanlisib (PI3Kα, δ-selective inhibitor) + ketogenic diet (insulin suppression) (NCT04750941)	II	Relapsed or refractory follicular lymphoma or relapsed or refractory EC with a documented activating mutation in *PIK3CA* or *PTEN* loss	Study status: Active, not recruiting Estimated study completion date: December 2024	
MEK inhibition				
Buparlisib (Pan-class I P13K inhibitor) + trametinib (MEK1/2 inhibitor), Bedard 2015 [81]	Ib	Advanced solid tumors with *RAS* or *BRAF* mutations (*n* = 113) including OC (*n*= 21)	G1/2: dermatitis acneiform, diarrhea, blood CK increased, stomatitis, nausea, rash macular, rash maculopapular, vomiting, AST increased, fatigue, ALT increased, dry skin, hyperglycemia, decreased appetite, asthenia, hypertension G3/4: blood CK increased	ORR 6.2%, OC 28.6%
Voxtalisib (Dual PI3K/mTOR inhibitor) + pimasertib (MEK1/2 inhibitor), Schram 2018 [82]	Ib	Advanced solid tumors with alteration in one or more of: *PTEN*, *BRAF*, *KRAS*, *NRAS*, *PIK3CA*, *EGFR*, *ERBB2*, *MET*, *RET*, *c-KIT*, *GNAQ*, *GNA11* (*n* = 146), OC (*n* =12)	G1/2 (>20%): diarrhea, fatigue, nausea, vomiting, dermatitis acneiform, maculopapular rash, peripheral edema, pyrexia, decreased appetite, stomatitis, dyspnea G3/4: hyponatremia, disease progression, hypokalemia, maculopapular rash, fatigue	ORR 4.1%, OC 8.3%
Uprosertib (Pan-AKT isoform inhibitor) + trametinib (MEK1/2 inhibitor), Westin 2019 [83]	I	Recurrent EC (*n* = 26)	G1/2: blood and lymphatic system disorders, eye disorders, gastrointestinal disorders, general disorders and administration site conditions, investigations, metabolism and nutrition disorders, musculoskeletal and connective tissue disorders, nervous system disorders, renal and urinary disorders, respiratory, thoracic, and mediastinal disorders, skin and subcutaneous tissues disorders, vascular disorders G3: blood and lymphatic system disorders, gastrointestinal disorders, general disorders and administration site conditions, metabolism and nutrition disorders, skin and subcutaneous tissues disorders. vascular disorders	ORR 3.8%
Voxtalisib (Dual PI3K/mTOR inhibitor) + pimasertib (MEK1/2 inhibitor), Arend 2020 [84]	II	Recurrent unresectable borderline/low malignant potential or LGSOC (*n* = 32)	G1/2/3/4: anemia, blurred vision, visual impairment, macular detachment, retinal detachment, diarrhea, nausea, stomatitis, dry mouth, vomiting, abdominal pain, fatigue, peripheral edema, chills, blood CPK increase, ALT increase, AST increase, arthralgia, myalgia, dizziness, paresthesia, dermatitis acneiform, alopecia, dry skin, maculopapular rash, pruritis, rash, cardiac disorders	ORR 9.4%
Buparlisib (Pan-class I P13K inhibitor) + binimetinib (MEK1/2 inhibitor), Bardia 2020 [85]	Ib	Advanced solid tumors with *KRAS*, *NRAS*, or *BRAF* mutations (*n* = 89) including OC (*n* = 19)	G1/2: blood CPK increased, diarrhea, AST increase, stomatitis, ALT increase, nausea, maculopapular rash, rash, dermatitis acneiform, fatigue, vomiting, decreased appetite, amylase increased, hyperglycemia, chorioretinopathy, lipase increased, peripheral edema, pruritus, dry skin G3/4: blood CPK increased, AST increase, ALT increase, maculopapular rash	ORR 12%, OC 27.8%
Everolimus (mTORC1 inhibitor) + avutometinib (RAF/MEK inhibitor) (NCT02407509)	I	Advanced solid tumors or multiple myeloma with RAS/RAF/MEK pathway mutations including *BRAF*, *KRAS* and *NRAS*	Study status: Recruiting Estimated study completion date: May 2024	
Endocrine therapy				
Everolimus (mTORC1 inhibitor) + anastrozole (aromatase inhibitor), Wheler 2014 [86]	I	Advanced HR+ gynecological or breast cancers (*n* = 55) including OC (*n* = 10), EC (*n* = 6), CC (*n* = 2) Expansion cohort: *PTEN* loss/*PIK3CA* mutations (*n* = 22)	G1/2: mucositis	ORR 10%, OC 0%, EC 0%, CC 0%
Everolimus (mTORC1 inhibitor) + letrozole (aromatase inhibitor), Colon-Otero 2017 [87]	II	Relapsed ER+ HGSOC (*n* = 20)	G1/2: exact percentages not reported G3/4: abdominal pain, anemia, small bowel obstruction, neutrophil count decreased, white blood cell count decreased, mucositis oral	ORR 16%
Everolimus (mTORC1 inhibitor) + letrozole (aromatase inhibitor), Slomovitz 2022 [88]	II	Persistent or recurrent EC (*n* = 37)	G1/2: exact percentages not reported G3/4: anemia, hyperglycemia	ORR 22%
Vistusertib (Dual mTORC1/2 inhibitor) + anastrozole (aromatase inhibitor), Heudel 2022 [89]	I/II	HR+ recurrent or metastatic EC (*n* = 49)	G1/2: nausea, fatigue, vomiting, diarrhea, arthralgia, decreased in lymphocytes count, hyperglycemia, anemia G3/4: decrease in lymphocytes count, hyperglycemia	ORR 24.5%
Everolimus (mTORC1 inhibitor) + levonorgestrel IUD (progesterone) (NCT02397083)	II	Complex atypical hyperplasia and stage IA grade 1 EC	Study status: Active, not recruiting Estimated study completion date: September 2026	
Inavolisib (PI3Kα-selective inhibitor) + letrozole (aromatase inhibitor) (NCT04486352) (EndoMAP)	Ib/II	Recurrent or persistent EC	Study status: Recruiting Estimated study completion date: October 2026	
Copanlisib (PI3Kα, δ-selective inhibitor) + fulvestrant (estrogen receptor antagonist) (NCT05082025)	II	ER+ and/or PR+ OC, EC, and breast cancers with PI3K (*PIK3CA*, *PIK3R1*) and/or *PTEN* alterations	Study status: Active, not recruiting Estimated study completion date: August 2024	
Ipatasertib (Pan-AKT inhibitor) + megestrol acetate (progestin) (NCT05538897)	Ib/II	Recurrent or metastatic endometrioid EC	Study status: Recruiting Estimated study completion date: January 2027	
Cell cycle checkpoint inhibition				
Samotolisib (Dual pan-class I PI3K/mTORC1/2 inhibitor) + prexasertib (CHK1 inhibitor), Hong 2021 [90]	Ib	Advanced solid tumors (*n* = 53) including EC (*n* = 3), CC (*n* = 1)	G1/2 (≥20%): white blood cell count decreased/neutrophil count decreased, platelet count decreased, nausea, anemia, vomiting, fatigue, diarrhea, stomatitis, rash, decreased appetite G3/4: white blood cell count decreased/neutrophil count decreased, platelet count decreased, anemia, rash	ORR 15.1%, EC 33%, CC 0%
Copanlisib (PI3Kα, δ-selective inhibitor) + elimusertib (ATR inhibitor) (NCT05010096) (COPABAY)	I	Advanced solid tumors with a defect in one or more DDR genes	Study status: Withdrawn by company sponsor secondary to no participants enrolled	
Gedatolisib (Dual PI3K/mTOR inhibitor) + palbociclib (CDK4/6 inhibitor) (NCT03065062)	I	Advanced squamous cell lung cancer, advanced pancreatic cancer, advanced head and neck cancer, or any tumor with suspected PI3K-pathway dependence (such as EC)	Study status: Recruiting Estimated study completion date: January 2026	
Triple pathway combination				
Everolimus (mTORC1 inhibitor) + letrozole (aromatase inhibitor) + metformin (biguanide derivative), Soliman 2020 [91]	II	Advanced or recurrent endometrioid EC with *KRAS* mutation in at least 18 patients (*n* = 62)	G1/2: exact percentages not reported G3/4: anemia, hypertriglyceridemia	ORR 28%
Serabelisib (PI3Kα-selective inhibitor) + nab-paclitaxel (microtubule-stabilizing agent) + insulin suppressing diet (insulin suppression) (NCT05300048)	I	Advanced solid tumors with *PIK3CA* mutations with or without *PTEN* loss	Study status: Recruiting Estimated study completion date: September 2024	
Everolimus (mTORC1 inhibitor) + letrozole (aromatase inhibitor) + metformin (biguanide derivative) (NCT01797523)	II	Advanced or recurrent EC	Study status: Active, not recruiting Estimated study completion date: October 2025	
Everolimus (mTORC1 inhibitor) + trametinib (MEK1/2 inhibitor) + lenvatinib (VEGFR-1, -2, -3, FGFR-1, -2, -3, PDGFR-α, RET, c-Kit inhibitor) (NCT04803318)	II	Recurrent/refractory advanced solid tumors	Study status: Recruiting Estimated study completion date: January 2027	
Everolimus (mTORC1 inhibitor) + letrozole (aromatase inhibitor) + ribociclib (CDK4/6 inhibitor) (NCT03008408)	II	Advanced or recurrent EC	Study status: Active, not recruiting Estimated study completion date: August 2028	

Abbreviations: AE = adverse effect, AUC = area under curve, BID = twice daily, CC = cervical cancer, DDR = DNA damage response, EC = endometrial cancer, ER = estrogen receptor, FC = fallopian tube cancer, gyn = gynecological, G3/4 = grade 3 or 4, HGSOC = high grade serous ovarian cancer, HR = hormone receptor, LGSOC = low grade serous ovarian cancer mg = milligram, OC = ovarian cancer, ORR = overall response rate, PC = primary peritoneal cancer, PO = oral, PR = progesterone receptor, QD = once daily, QOD = every other day, QW = once weekly, Q2W = once every two weeks, Q3W = once every three weeks, VC = vaginal cancer, 21/7 = 21 days-on, 7 days-off.

## Data Availability

No new data were created or analyzed in this study. Data sharing is not applicable to this article.

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
