# Peer review of "Combination Therapy Approach to Overcome the Resistance to PI3K Pathway Inhibitors in Gynecological Cancers"

_cells, 2024, doi:10.3390/cells13121064_

Round 1
Reviewer 1 Report
Comments and Suggestions for Authors
The article by Ibanez et al. is a comprehensive revew on PI3K inhibitors in gynecologic cancers. It is well written and well up to date on the subject. Nevertheless, one very recent study with combination therapy might deserve a notion in this respect:
Combination treatment with alpelisib and olaparib indicated a negative benefit-risk as it did not improve progression-free survival (PFS) over physician’s choice of treatment in a phase 3 trial of patients with platinum-resistant or -refractory high-grade serous ovarian cancer
Results from this trial, EPIK-O, were presented at the SGO 2024 Annual Meeting on Women’s Cancer.
Author Response
Thank you for your positive feedback and for highlighting the EPIK-O trial results. We have discussed it in the Conclusion and future directions section (lines 502-505).
Reviewer 2 Report
Comments and Suggestions for Authors
In this review article, the authors provided an overview and current advance of PI3K pathway inhibitors in gynecological cancer treatment, described resistance mechanisms against PI3K pathway inhibitors and summarized recent clinical trials that combine PI3K pathway inhibitors with various other pathway inhibitors to overcome the resistance. This manuscript is well written, easy to follow and would be an important source in understanding the use of PI3K pathway inhibitors in gynecological cancer patients. However, there are some questions in Figure 1 and Table 1, which need to be answered.
Figure 1: Please double check this figure to ensure all the information is accurate.
1. Some inhibitors are mentioned in Table 1, but not listed in Figure 1
2. The activation of AKT by mTORC2 is not shown in Figure 1. The arrow shown here is only from AKT to mTORC2, indicating the activation of mTORC2 by AKT.
3. The activation of mTORC1 by AKT is via inhibition of TSC1/2. TSC1/2 should be a negative regulator in Figure 1.
Table 1
Duvelisib is listed twice in this Table.
Author Response
Figure 1: Please double check this figure to ensure all the information is accurate.
1. Some inhibitors are mentioned in Table 1, but not listed in Figure 1
Response: All inhibitors have been listed in Figure 1.
2. The activation of AKT by mTORC2 is not shown in Figure 1. The arrow shown here is only from AKT to mTORC2, indicating the activation of mTORC2 by AKT.
Response: We have added a double-headed arrow between AKT and mTORC2 in Figure 1.
3. The activation of mTORC1 by AKT is via inhibition of TSC1/2. TSC1/2 should be a negative regulator in Figure 1.
Response: TSC1/2 have been corrected as negative regulators.
Table 1: Duvelisib is listed twice in this Table.
Response: The duplicate entry for duvelisib has been removed from Table 1.
Reviewer 3 Report
Comments and Suggestions for Authors
A timely review article by Dr. Huang and the group elaborates on the mechanisms and therapeutic approaches to overcoming resistance to PI3K inhibitors. This is a very well-written and informative review article with a translational aspect. A few things need to be addressed before it is ready for acceptance. They are as follows:
1. It is well known that the Hippo pathway integrates PI3K-Akt signals (PMID: 23359693), and it has recently been shown that YAP1/TAZ rescue cancer cells from KRASG12C inhibition by reactivating the functions of ERK-dependent transcription factors and upregulating PI3K/AKT activation (PMID: 37729426). This clearly suggests that combining PI3K, KRAS, and YAP/TEAD inhibitors would be more efficacious and effective as a combination therapy to combat resistance. This topic must be addressed as one of the proposed future aspects of using PI3K inhibitors as combination therapy. Add a few lines discussing this by referring to the abovementioned relevant work.
2. Another novel approach would be combing AMPK activator along with PI3K inhibitors. It has been shown that mTOR inhibitor can synergistically efficacious when combined with AMPK activators like metformin (PMID: 26323019), and also been shown that Metformin induces apoptosis and inhibits migration by activating the AMPK/p53 axis and suppressing PI3K/AKT signaling (PMID: 30533337).
This should be discussed in a few lines as a perspective using PI3K inhibitors as novel combo therapy. Add a few lines discussing this by referring to the abovementioned relevant work.
3. Figure 3, Add SHP2 next to the GAB2, GRB2 and SOS. For references, use PMID: 37682219.
Author Response
1. It is well known that the Hippo pathway integrates PI3K-Akt signals (PMID: 23359693), and it has recently been shown that YAP1/TAZ rescue cancer cells from KRASG12C inhibition by reactivating the functions of ERK-dependent transcription factors and upregulating PI3K/AKT activation (PMID: 37729426). This clearly suggests that combining PI3K, KRAS, and YAP/TEAD inhibitors would be more efficacious and effective as a combination therapy to combat resistance. This topic must be addressed as one of the proposed future aspects of using PI3K inhibitors as combination therapy. Add a few lines discussing this by referring to the abovementioned relevant work.
Response: We acknowledge the interaction between the Hippo/YAP1 and PI3K/AKT pathways (PMID: 23359693) and its role in resistance to KRASG12C inhibition in non-small cell lung cancer (PMID: 37729426). After carefully reviewing the suggested references and relevant literature, we have included a reference supporting the role of YAP1 in AKT inhibitor resistance in ovarian and endometrial cancer cell lines [1] and referred the suggested reference for combining PI3K, KRAS, and YAP/TEAD inhibitors (PMID: 37729426) as a future therapeutic strategy in the Conclusion and future directions section (lines 496-502).
2. Another novel approach would be combing AMPK activator along with PI3K inhibitors. It has been shown that mTOR inhibitor can synergistically efficacious when combined with AMPK activators like metformin (PMID: 26323019), and also been shown that Metformin induces apoptosis and inhibits migration by activating the AMPK/p53 axis and suppressing PI3K/AKT signaling (PMID: 30533337).
This should be discussed in a few lines as a perspective using PI3K inhibitors as novel combo therapy. Add a few lines discussing this by referring to the abovementioned relevant work.
Response: We have included relevant discussions on AMPK activator metformin with mTOR inhibitors in endometrial cancer (references 36, 74, 90, 96, lines 388-401). Thus, we believe additional references on breast (PMID: 26323019) and prostate cancer cells (PMID: 30533337) are unnecessary.
3. Figure 3, Add SHP2 next to the GAB2, GRB2 and SOS. For references, use PMID: 37682219.
Response: We have updated Figure 3 with the suggested reference (reference 40 in the manuscript).
Reference:
- Previs, R.A.; Armaiz-Pena, G.N.; Ivan, C.; Dalton, H.J.; Rupaimoole, R.; Hansen, J.M.; Lyons, Y.; Huang, J.; Haemmerle, M.; Wagner, M.J.; et al. Role of YAP1 as a Marker of Sensitivity to Dual AKT and P70S6K Inhibition in Ovarian and Uterine Malignancies. J Natl Cancer Inst 2017, 109, doi:10.1093/jnci/djw296.
Round 2
Reviewer 3 Report
Comments and Suggestions for Authors
All concerns addressed, ready for acceptance